# Enhancing Intersectoral Collaboration in Maternal Healthcare for the Realization of Universal Health Coverage in Kenya: The Perspectives of Health Facility Administrators in Kilifi County, Kenya

**DOI:** 10.3390/ijerph22040610

**Published:** 2025-04-14

**Authors:** Stephen Okumu Ombere

**Affiliations:** Department of Sociology and Anthropology, Maseno University, Maseno 40109, Kenya; sokumu2@gmail.com

**Keywords:** intersectoral collaboration, Kilifi County, qualitative research, maternal health, universal healthcare

## Abstract

Intersectoral collaboration is an instrument that enables better productivity by filling in for possible gaps in knowledge, skills, and competencies in a given department by leveraging them from other departments. In Kenya, there is a paucity of information on intersectoral collaboration in healthcare. This article explores the possibilities of intersectoral collaboration, specifically in maternal healthcare, and what can be done to realize such collaborations to drive universal health coverage (UHC) in Kenya. Free maternity services (FMSs) are among the primary healthcare services that push Kenya towards UHC. In light of the centrality of UHC in driving current health policy, there are still several challenges which must be faced before this goal can be achieved. Moreover, competing priorities in health systems necessitate difficult choices regarding which health actions and investments to fund; these are complex, value-based, and highly political decisions. Therefore, the primary objective of this article is to explore health facility administrators’ views on whether intersectoral collaboration could help with the realization of UHC in Kenya. The study area was Kilifi County, Kenya. The article is based on follow-up qualitative research conducted between March and July 2016 and from January to July 2017, and follow-up interviews conducted during COVID-19 in 2020 and 2021. The data are analyzed through a thematic analysis approach. The findings indicate that through *Linda Mama*, the expanded free maternity services program is one of the possible pathways to UHC. However, participants noted fair representation of stakeholders, distributed leadership, and local participation, considering bargaining power as a key issue that could enhance the realization of UHC in intersectoral collaboration through *Linda Mama*. These techniques require a bottom–up strategy to establish accountability, a sense of ownership, and trust, which are essential for UHC.

## 1. Introduction

Health is a requirement, contributor, and clear pointer for sustainable development goals (SDGs) and is core to propelling the 2030 Agenda [1,2]. In the SDGs, there is only one goal which explicitly refers to health (SDG 3). Moreover, some targets are important for health improvement across the other 16 SDGs [3]. The majority of the health SDG targets, like reducing avoidable child deaths and the HIV epidemic, are too big for the health sector to handle on its own. Under other goals, they rely on the SDG targets being met. Thus, the SDGs bring attention back to the ways in which intersectoral action can be used to enhance health outcomes and the ways in which the health sector can collaborate with other sectors to accomplish the full range of SDG targets, not just those focused on SDG 3 [4,5,6]. Many states have crafted policies specifically addressing universal health coverage (UHC), while others have merged UHC goals into their broader healthcare sector approaches. This extensive adoption shows that there is a growing acknowledgment of the relevance of UHC in enhancing population health and reducing catastrophic healthcare costs [7,8].

In Kenya today, UHC appears to represent a new approach and new way of thinking about poverty and redistribution, the state and citizenship, health and development, as well as an understanding of the necessity for innovative approaches to address injustice and inequality [9]. UHC thus represents a novel strategy and fresh perspectives on the state and citizenship, health, and development in the twenty-first century, and on poverty and redistribution [10]. Moreover, Prince, 2017 [10] argues that UHC is unique in two ways: first, in terms of its goals for social solidarity; and second, in its understanding of the state’s pivotal role in social protection as it relates to health [11]. Kenya has advanced toward UHC, as seen by the numerous reforms and policy measures that have been implemented since the country gained independence [4,12]. UHC has two primary goals: improve financial risk protection and intensify access to necessary care for citizens [3,9]. According to World Health Organization, 2015 [13], achieving these goals requires proper prioritization and budgeting from governments and necessitates offering financial risk protection between the wealthiest and poorest, the healthy, and the ill [14].

The attention of the health sector in many countries remains entirely on healthcare services, and the potential for intersectoral collaboration remains untapped in many low- and middle-income countries [5]. Although recent studies explored the various health components that can propel the UHC agenda in Tanzania [15] and Kenya, as reported by healthcare professionals, patients, and other key policymakers [12,15,16], these studies did not explore whether intersectoral collaborations can lead to the realization of UHC. Studies have also explored the relevance of engagement of different stakeholders, including the community, and advocated for bottom–up approaches in pushing for health policy reforms in other contexts [17,18,19,20,21]. These studies did not explore the possibilities of enhancing intersectoral collaborations for healthcare. Moreover, Muinde and Prince, 2022 [9] argue that additional research is needed to better understand the silos that exist in the creation of health policy, especially when it comes to implementing these approaches. Moreover, in Kenya, there is no clear description of intersectoral collaborations in the current maternal healthcare policy. This article, therefore, explores the perspectives of healthcare administrators to ascertain whether it is possible to establish an intersectoral collaboration to help Kenya realize UHC through maternal healthcare policy.

Greater support is needed for countries to embark on intersectoral health action in the wake of the SDGs, including establishing the health sector’s engagement in this action as an integral part of the universal healthcare (UHC) agenda [5,22,23]. Existing efforts on this theme—for example, those based around the “Health in All Policies” approach and on the political economy of intersectoral action for health—can be drawn upon to support several countries in doing so [24]. Intentional cooperation between different sectors (such as the economy, health, and environment) and stakeholder groups (such as the government, civil society, and the private sector) to jointly achieve a policy objective is referred to as an intersectoral (or multisectoral) strategy. In order to improve health outcomes, partners can make the most of their combined and diverse assets by involving several sectors and leveraging their knowledge, skills, reach, and resources. It is not easy to improve public health (PH) due to the large population and geographical variations [25,26].

Kenya has made substantial improvements in the accessibility of health services and significantly decreased its mortality rates over the last twenty years [27]. Even though Kenya has successfully lowered maternal mortality, there are still significant disparities in the country’s access to healthcare and health outcomes [27]. Kenya is among the many African nations dedicated to pushing for health system shifts through the provision of fair and reasonably priced access to vital medical treatments. Facility-based deliveries increased in Kenya from 44% in 2008 to 61% in 2015 [28]. A portion of the increase has been associated with the June 2013 implementation of the free maternity services (FMSs) policy [29]. Research indicates that, despite this surge in facility delivery, the FMSs policy was hastily enacted out of political expediency and lacked public and stakeholder participation. This impeded the effective execution of the policy by various parties [30,31,32].

Healthcare services in Kenya are provided by public health hospitals, private-for-profit facilities and non-governmental organizations. Public health facilities are organized around a four-level system: (1) community services, (2) primary health services, (3) county referral services, and (4) national referral services [33,34]. Although both private and public facilities charge user fees to clients, these fees are subsidized at level one and two in public hospitals. The private facilities are more expensive compared to public ones, primarily serving wealthier individuals, whereas those from poorer households more commonly rely on public care providers or use lower-standard private care facilities [35]. County governments are responsible for providing services on levels one to three, and the national government is responsible for providing national referral services [36]. In addition, under the new framework, responsibility for health service delivery is assigned to the counties, whereas policy, national referral hospitals, and capacity building are the national government’s responsibility [36]. Moreover, Kenya has enshrined the right to health in Article 43 (1) (a) of the Constitution of Kenya, 2010, which states that “Every person has the right to the highest attainable standard of health, which includes the right to healthcare services, including reproductive health” [37]. Additionally, Kenya’s Vision 2030, a long-term development plan, acknowledges the importance of sexual and reproductive health rights (SRHRs) as a key component of improving the quality of life for all Kenyans, with particularly focus on adolescents and young people. This is reflected in the Social Pillar of Vision 2030, which aims to provide equitable, affordable, and quality healthcare, including SRHR services through UHC [12,37].

Research suggests that applying the principle of participation at all policy formulation and implementation stages in sub-Saharan Africa can help address the underlying causes of public health concerns and create long-term, culturally appropriate remedies [38]. Recently, Muinde and Prince, 2022 [9] explored the finding that recent government initiatives to increase access to healthcare, such as providing free healthcare services and changing national health insurance coverage, were perceived differently by “ordinary citizens”, or *wananchi*. Despite these improvements, the healthcare system remained historically divided due to class disparities, patronage politics, and exclusionary practices that all opposed universal access. Promises of inclusion concealed realities of exclusion, drawing people to entrenched forms of failure, neglect, and difference. They began to question the government’s obligation, solidarity, their rights to healthcare, and the rise in class disparity in accessing healthcare. This article sheds light on whether disparities and trust can be requisite in intersectoral collaborations in a healthcare system. Thus, the primary objective of this article is to explore health facility administrators’ views on whether intersectoral collaboration could help with the realization of UHC in Kenya.

## 2. Methods

### 2.1. Study Setting

This qualitative research was carried out in Kenya’s coastal region, Kilifi County. Kilifi is categorized as a semi-arid and arid region. Seasonal water shortages affect more than 65% of Kilifi people, and floods and droughts impact food security and production. It has a high rate of poverty—66.7%—and pervasive food insecurity, which affects roughly 67% of households. A large proportion of people live in rural areas [28]. The dominant community is the Giriama sub-tribe within the larger MijiKenda population. The Giriama people mostly depend on subsistence farming, with wage work in salt mines, small-scale trading, cashew nut production, palm wine production, and animal husbandry serving as additional sources of income. Kilifi is among the top 15 counties in Kenya that contribute to maternal and perinatal deaths [21,39]. The study was conducted in Kilifi County referral hospital, as it is the main referral hospital in the county. The study area is also explicitly described in the author’s previous article [12,30,40,41].

### 2.2. Study Design

The study design was qualitative. This research was an integral part of the larger project on inclusive growth through social protection in maternal health programs in Kenya (SPIKE), which was a multidisciplinary research project. This article is derived from the larger goal of this study, which aimed at understanding local opinions of social protection programs. One of the major themes of this study is local participation during policy crafting [11]. The study explored health facility administrators’ views on whether intersectoral collaboration could help in the realization of UHC in Kenya. The study was longitudinal.

### 2.3. Study Sample and Methods of Data Collection

Data collection was longitudinal with follow-up interviews. Therefore, the research was performed from March to July of 2016 and also from January to July of 2017. Moreover, the researcher conducted follow-up interviews with health administrators during the COVID-19 pandemic 2020 conducted between June and July 2020 and September and October 2020. Health administrators were purposively selected. The inclusion criteria included key healthcare officials dealing with maternal and child health, and also facilitating health issues in the hospital and at the county level. They were required to have worked in administrative positions for more than three years before the onset of the study. Lower-cadre healthcare workers and those who did not work in an administrative position were excluded from this study. The study utilized a total of 10 key informant interviews with health administrators. They were concerned with the facilitation of healthcare issues at the hospital and county level. Therefore, their viewpoint on the introduction and the implementation of free maternity services and whether they were a roadmap to UHC was insightful. The researcher also conducted two round-table discussions with county health administrators at the county health boardroom. Each round-table discussion had four health administrators as the participants. During the discussion, notes were taken and the discussion recorded using an audio-voice recorder. In 2016 and 2017, the researcher conducted informal conversations with the health administrators within the health facility. The interviews explored local perceptions of social protection schemes in maternal health in Kenya. Additionally, the researcher purposefully picked all five matrons—nurse–midwives who oversee departmental operations—during the initial fieldwork and only the two busiest health facilities within the county were included. During the COVID-19 pandemic in 2020, the researcher conducted follow-up interviews. The five matrons that had initially been interviewed formed the study population, as described in the author’s previous study [39]. Thus, the study population in this study was the matrons and the health administrators in the county.

### 2.4. Data Analysis

The author utilized thematic analysis to analyze the qualitative data in this study [42]. The researcher utilized a computer program known as F5 transcription-free to perform the transcriptions. Additionally, the author’s handwritten notes from informal conversations were also manually coded and the themes were added to the study. The author and his supervisors assessed the codes to identify recurring themes throughout the study. Data analysis stopped when no new themes were emerging. The results are described comprehensively in the text and are supported by exact quotes.

### 2.5. Ethical Procedure

The objective of the study was made known to the research participants. They had the option to stop participating in the study at any point throughout the interviews; participation was entirely at their discretion. All participants were informed about the nature of this study, and only those who consented to the phone interviews participated. No participants declined to participate in the study. Ethical approval was attained from Maseno University Ethical Review Committee, reference number MSU/DRPI/MUERC/00206/015, date of approval: 2 March 2017.

## 3. Findings

This study’s findings revealed several themes that help explain healthcare workers’ viewpoints on the possibilities of intersectoral collaborations for attaining UHC in Kenya. These themes include fair representation of stakeholders, the need for collaborative and distributed leadership, and local participation and bargaining power.

### 3.1. Representation of Stakeholders

In an interview with a maternity and child health clinic nurse and a round-table discussion, the participants expressed the significance of having equal representation in the development and application of health policies. The participants contended that inadequate representation was a significant disadvantage of using FMSs as a social protection program. A verbatim quote from the nurse shows that intersectoral collaboration may be possible, but there ought to be a clear representation of stakeholders.


*FMS is a good social protection program. But, if healthcare providers, especially those charged with maternal and child health, are not included during policy making, it is challenging to implement. But there should be an even representation, which should include local people, the local administration, and other social institutions that can help us push the FMS agenda. We neglected community health volunteers in this policy, but they play a critical role in healthcare indicators. So, UHC is still an uphill task, and we must collaborate to drive the UHC agenda (Interview with health facility nurse).*



*FMS has a drawback: lack of presentation. We cannot pretend that UHC can be achieved without intersectoral collaboration. However, such collaborations should have all stakeholders dealing with health on board. Such collaborations should consider fair representation in the policy implementation. Healthcare is not a stand-alone entity when we want to achieve goals such as UHC (Roundtable discussant).*


Moreover, during the round-table discussions, it emerged that there is a need to include relevant institutions to propel the UHC agenda. This meant that there is a need for a collaborative approach. The participants noted that such collaborations should be an ongoing continuous process. The participants also said that such a continuous process would make the local population and other actors aware of the viability and relevance of FMSs in the realization of UHC. Thus, a collaborative approach will enhance a bottom–up initiative that will enable institutions involved in the UHC agenda to have more realistically localized control over resources from the national government through active representation.


*If we have representation and continuous bottom-up engagement with various actors, then UHC agenda will never be doubted. Again, nobody will allow people to play around with resources meant to help the mothers because when the people and institutions are involved, they have a sense of ownership of such policies. Community involvement should be a continuous process. It should include educating the people (An informal conversation with the health facility administrator).*


A health worker added that


*If at all, we engage community members continuously…for instance, we have involved community health volunteers during maternity health talks. Today, most mothers know their right to maternity and have changed their attitude toward health facility delivery courtesy of free maternity (Interview, Matron 06 in-Charge MCH, Kilifi county).*


### 3.2. Distributive Leadership

The participants suggested that distributive leadership is key to effective governance of intersectoral action. Building competencies in leadership across government levels and sectors is necessary, as is raising up supporters in various domains who can come to a resolution regarding shared objectives. Otherwise, relying on national and county governments alone is inadequate for intersectoral healthcare collaborations. Participants suggested that leaders with expertise in different aspects of maternal healthcare policy can team up to enhance intersectoral collaboration.


*We have different organizations with leadership skills. We have experts in mobilization, community sensitization and even policy interpretation to the people. We need to bring such actors on board. I tell you, even for mothers in the villages to appreciate the government’s initiative on FMS. We must get every leader and expert from different fields on board (Interview with Health administrator).*



*We can learn from the leadership of other organizations that do not necessarily deal with health issues. We can develop a clear implementation agenda and strategy, which can enhance transparency. FMS is here to stay, but we need to create and proactively push for UHC through the services women get in this hospital. Even the UHC agenda must address how health facilities can work with other sectors. The agenda should be flexible (round-table discussion with county health officials).*


The participants stated that conversations and policies about the path to UHC should incorporate people’s most important needs by incorporating leadership from different organizations. The participants also suggested some crucial ways to support the county and national government in pushing for an intersectoral approach to health. This includes ensuring that the UHC agenda considers the health sector’s ability to collaborate with other sectors. Moreover, allowing other leaders to come on board for knowledge sharing purposes was also mentioned.


*Mothers say they were never consulted, and even health workers were never consulted. I wish the county and national governments consulted wider. Apart from the politicians, they can bring other leaders and experts from the community, local NGOs, and other partners to help drive the UHC agenda home (FGD participant, April 2017).*


### 3.3. Local Participation and Bargaining Power

It also became clear that, in order to achieve UHC in Kenya, community involvement is essential to local, equitable, and integrated healthcare. Community members must be seen as more than just passive users of healthcare in policies and initiatives to enhance primary healthcare. Rather, they ought to be leaders who play a significant part in organizing, making decisions, and conducting evaluations. Involvement of all communities is a requisite for *Linda Mama* to achieve its objectives. The participants noted that the engagement of citizens is key to developing UHC schemes that are responsive to people’s needs and priorities. Moreover, the participants argued that engaging health providers in decision-making gives them more bargaining power.


*Obstacles exist for free maternity care, and some community members oppose hospital deliveries. After all, the people are facing economic hardships so free maternity care is still costly for them. Not even Linda Mama covers the costs of X-rays, medications, lab testing, or transportation to a medical institution. However, we sure must ensure that women safely give birth. Even if we believed that the government would improve the welfare of healthcare providers, they had to go on strike for them to be heard. Additionally, our community has to be included from the beginning (round-table discussion with authorities from the county health team).*



*Local participation, or let me just say public participation, is very important when we want to have a successful policy rollout. Such participation will give people a voice to defend UHC and to demand services when they visit the hospital. For instance, people here helped in the acceptance of the rubella vaccine to women in the community, because we involved them and employed the local community health volunteers. Thus, such participation and involvement can also yield good results in FMS (an interview with an administrator).*



*If we can bargain well with the national government on what we want, I believe Linda Mama will be one of the perfect avenues to push the UHC agenda in Kenya. But here we are: We don’t have the room to bargain with the government at that level. We get policy documents to implement what is needed. I am looking forward to the day health providers will be consulted before a policy is rolled out. When consulted, we have higher bargaining power and cannot accept intimidation (an interview with a health provider).*


## 4. Discussion

This section discusses key themes from the varying views of health facility administrators and county health officials regarding enhancing intersectoral collaborations in maternal healthcare to realize universal health coverage in Kenya.

Any activity involving non-health sectors that has the potential to promote health is considered an intersectoral or multisectoral health action [5]. Findings show that a fair and robust representation of stakeholders will enable the expanded free maternity services to push forward the UHC agenda, accelerating the creation of more accountable and participatory institutions. As a result, all interested parties’ right to representation from the community needs to be included in the policy discussion. Pradier et al., 2023; Rifkin, 2014 [43,44] observed that identifying a feasible solution and/or its acceptance in the community may be jeopardized if an essential stakeholder is not involved. Moreover, this study’s findings corroborate various studies [38,45,46,47], which indicate that community participation and involvement of different stakeholders make it more likely that a policy will effectively adapt to the local circumstances. Additionally, Obi, 2024 [20] also argues that for the realization of UHC, strategic involvements and collaborations of local actors within communities will lead to improvements in the utilization of primary healthcare centers and that community members will also be granted increasing access to maternal healthcare services.

Furthermore, reports by Pratt, 2018; 2019 [48,49] demonstrate how representation improves the adoption of health policies and is a necessary condition for such participation. This suggests that, in contrast to how the policy was implemented without fair representation, having community representation steer the policy’s formulation will increase the understanding of and positive reactions toward FMSs. Furthermore, based on the study’s findings, I contend that the SDGs are not the only factor necessary for Kenya to achieve UHC. It could be challenging to determine the appropriate course of action from a single sectoral (health, education, agriculture, environmental sustainability, etc.) perspective due to the interconnectivity of the SDGs. Bennett et al., 2020 [50] argue that the SDGs’ paradigmatic shift pushes for unique thinking to address the interconnected SDGs. One sure way of addressing the interconnected nature of SDGs is through fair representation of stakeholders’ rights from the grassroot level. From this study, it is clear that long-term benefits can result from bringing together representatives of many communities, organizations, disciplines, backgrounds, and cultures to share information, debate evidence, and offer solutions for the ongoing potential achievement of UHC through *Linda Mama*.

In this study, it also emerged that collaborative leadership is an alternative approach to increasing local responsibility and engagement with society in the implementation of *Linda Mama* to promote transparency, inclusiveness, and participation. This suggests that more work is needed to develop either people-centered health systems or systems that address the needs of individuals and communities comprehensively. Such collaborations can be nurtured by addressing conflicting aims of power dynamics and fostering trust. Abbas et al., 2022; Fourie, 2001; Ombere et al., 2023 [17,51,52] asserted that when regulations are codified, the ability to influence healthcare policies can be removed from a small group of powerful people, giving the disadvantaged a voice and, ultimately, guaranteeing long-term representation. Moreover, despite the Kenya Vision 2030 social pillar’s emphasis on investing in people’s well-being, with health as a key sector, findings from this study show that affordable and accessible healthcare can also be made possible through intersectoral collaborations. Such collaborations could help make healthcare services affordable and accessible to all citizens, regardless of their socio-economic status. Therefore, collaborative, accountable, and distributive leadership in *Linda Mama* can propel Kenya to realize UHC and Vision 2030.

This study shows that there was a lack of local participation during the expansion of free maternity services. Health providers felt they had low bargaining power to push some agendas. They echoed that some agendas could not be achieved unless the community members were incentivized. Studies have also shown that the free maternity services program in Kenya was launched in a hurry, and it has been used as a pathway to universal health coverage and also considered a political challenge agenda [12,30]. In this study, I argue that to ascertain the features and extent of interventions, as well as their acceptance and eventual sustainability, stakeholder involvement in their design is crucial. Without this, even the most meticulously planned efforts will fail to provide the intended results. Research has revealed that many public health initiatives have focused on educating the public under the premise that knowledge is sufficient to alter behavior [43]. Local participation as a component of intersectoral collaboration enhances the possibility that a public health strategy will be appropriate from a cultural and educational standpoint, and that its structure and content will better suit the community’s cultural systems [12,45,47]. Research indicates that adapting the principle of participation at all policy crafting and implementation levels will assist in addressing the underlying causes of public health issues in sub-Saharan Africa and locating long-term, socio-culturally suitable remedies, particularly for UHC [20,38]. This study’s findings also reverberate with those of [52], who imply that the process of creating institutions requires “trust.” In addition to the care they receive while sick, people respect health systems for contributing to society’s overall well-being [12,53]. Moreover, community involvement depends on the level of linked and targeted activities for health engaged in by community members [20].

Therefore, for UHC to succeed through *Linda Mama*, local communities’ primary healthcare needs need to be known to legislators. In such a method, emic views would be necessary. Local confidence in politically linked programs and those locally owned through non-political means would be strengthened by allowing the locals to monitor *Linda Mama*. The results further support the World Health Organization’s suggestions that, to improve mother and child health (MCH) outcomes, active community participation should be encouraged during the planning and execution of programs [13].

## 5. Conclusions

Health equity is a complex, intersectoral issue requiring participation from multiple stakeholders. Despite the difficulties of implementing universal health coverage in Kenya, there is an increasing acknowledgement of its worth in improving access to healthcare. This study is a follow-up study on implementing the expanded free maternity services in Kenya dubbed *Linda Mama*. Specifically, the study shows the perspectives of health administrators regarding how *Linda Mama* can be a possible pathway to Kenya’s realization of UHC and whether intersectoral collaboration could work for UHC realization. Utilizing qualitative methods, it emerged that *Linda Mama* is a good initiative. Given the intertwined nature of the SDGs, and to propel Kenya to UHC, there is a need for intersectoral collaboration. To realize intersectoral collaboration, the participants in this study argued that there is a need for fair representation of stakeholders; collaborative, accountable, and distributive leadership; and local participation and bargaining power. Such approaches to intersectoral collaboration will provide a more holistic, sophisticated, and interrelated experience with regard to *Linda Mama’s* implementation, which would foster local ownership and advance Kenya’s sustainable fulfillment of UHC.

## 6. Limitations

The main strength of this study is that this is one of the first anthropological studies conducted among health administrators to give a primary account of what can be done to enhance intersectoral collaboration in the expanded free maternity program *Linda mama* for the realization of universal healthcare in Kenya. One of the major limitations is that the study only relied on health administrators’ perspectives, and since they are policymakers at the county, some of their responses might have been exaggerated. However, to counter this, this study utilized varying data collection methods to triangulate the findings, thus strengthening the findings. Moreover, the article lacks the perspectives of administrators from other institutions such as non-governmental organizations, faith-based organizations, private hospitals, and international organizations dealing with maternal healthcare. This is a research gap that can be investigated in the future. While the findings may not be generalizable beyond Kilifi county because of the heterogeneity of the counties, this study identifies significant contextual factors that can be informative to policymakers as a guide to effective evidence-based interventions that can be adopted to strengthen the implementation of universal health coverage in Kenya.

## Data Availability

The data can be accessed by contacting the author Stephen Okumu Ombere (sokumu2gmail.com), who can share anonymized transcripts. However, due to their qualitative and personal nature, the data are not publicly available.

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
