# Peer review of "Enhancing Intersectoral Collaboration in Maternal Healthcare for the Realization of Universal Health Coverage in Kenya: The Perspectives of Health Facility Administrators in Kilifi County, Kenya"

_ijerph, 2025, doi:10.3390/ijerph22040610_

Round 1

Reviewer 1 Report

Comments and Suggestions for Authors

I would like to commend the author for presenting an insightful and well-structured study on the role of intersectoral collaboration in achieving Universal Health Coverage (UHC) through maternal healthcare services in Kenya. The manuscript highlights a crucial area of public health policy and provides valuable qualitative insights from healthcare administrators. The study is highly relevant, given the global push for UHC, and particularly significant for low- and middle-income countries like Kenya. see attached 

Author Response

Dear Reviewer,

Thank you so much for your comments. The manuscript has been improved further for clarity.

Reviewer 2 Report

Comments and Suggestions for Authors

Dear Author,

Below are the considerations for improving the writing of the manuscript.

Authorship:

The author states in the Funding section that “This article is based on my PhD dissertation” and in the Acknowledgments section presents the following information:
“I am also grateful to my PhD supervisors Prof. Dr. Tobias Haller (Institute of Social Anthropology, University of Bern, Switzerland) and Dr. Sonja Merten (Swiss Tropical and Public Health Institute, Switzerland).”
I request clarification on why the PhD supervisors are not listed as co-authors of the article.

Keywords:

The keywords should be aligned with the MeSH thesaurus.
The following terms were not found in MeSH: Bottom-up; free maternity services; Kilifi County.
Replace “universal health coverage” with “Universal Health Care.”
Replace “qualitative” with “Qualitative Research.”

Introduction:

In the sentence: “In Kenya today, UHC symbolizes a shift away from a health care system focused on the Market,” could you better explain what “Market” refers to?
See: “universal healthcare (UHC) agenda.”
In the fifth paragraph of the introduction, I suggest explaining how the current healthcare system in Kenya is organized in general (including both primary care and hospital care), with a more detailed focus on maternal care.
Is the right to health in Kenya a constitutional right?
For policymakers in Kenya, what is their concept of health?
Include a contextualization of the Vision 2030 document – https://vision2030.go.ke/about-vision-2030/

Objective:

The author needs to clearly define the objective of the study presented in this manuscript. Once defined, the objective should serve as the guiding thread of the article. The results should directly address this objective, and the conclusion should be strictly related to it and the data presented.

I suggest presenting the same objective at the end of the introduction and in the abstract. However, in my opinion, both are not well aligned with the study.
My suggested objective is:

  • Explore health facility administrators’ views on whether intersectoral collaboration could help in the realization of UHC in Kenya
    OR
  • Explain healthcare workers’ viewpoints on the possibilities of intersectoral collaborations in attaining UHC in Kenya

In the conclusion, the following statement appears:
“This study is a reflection on implementing the expanded free maternity services in Kenya dubbed Linda Mama.”

Methods:

The Methods section should be aligned with the COREQ checklist (See EQUATOR Network – https://www.equator-network.org/reporting-guidelines/coreq/). All items should be covered in the methods section.

Item 2.3

Study Sample:

This section requires substantial revisions to clearly describe who the study participants were and how they were recruited.

  • The author must clearly state who participated in the research described in this manuscript.
  • Who were the participants in the individual interviews?
  • Since this is a longitudinal study, did the same participants take part in both the first phase (July 2016 and January–July 2017) and the second phase (2020 and 2021)? Which months were covered in the second phase?
  • Who participated in the two round-table discussions? How were these participants selected?
  • Did the individuals who took part in the informal conversations give consent to participate in the research? Were they aware that the information from these conversations would be used in a scientific study? How was the data from these conversations recorded?

“The researcher purposively picked five matrons—nurse-midwives who oversee departments—during my first fieldwork and also selected the two busiest health facilities.”

  • What criteria guided this purposive selection?

“The five matrons that had initially been interviewed formed the study population as described in the author’s previous study.”

  • Are the matrons the study population for this research? This is unclear.

Methods of Data Collection:

  • Provide information about the interview and round-table discussion guides.
  • What tool was used to record the data collection?

2.4. Data Analysis:

  • “The conclusion of the data analysis was reached when no new themes emerged.”
    • Is this correct? Or was it the conclusion of data collection?

2.5. Ethical Procedure:

  • Ethical approval was obtained from Maseno University Ethical Review Committee, reference number MSU/DRPI/MUERC/00206/015, date of approval: March 2, 2017.
  • However, data collection started in July 2016, before ethical approval was granted. Could you clarify this issue?

Findings:

According to the COREQ guidelines, qualitative research results should be presented as an interpretation of the meanings attributed by the participants, supported by excerpts from participants’ statements that validate these interpretations.
Additionally, the study should include a characterization of the participants.
Thus, I suggest rewriting this section.

Discussion:

It is necessary to include a discussion related to the Vision 2030 document – https://vision2030.go.ke/about-vision-2030/

Regarding this citation: “Studies have also shown that the free maternity services programme in Kenya was hurriedly launched, and it has been used as a pathway to universal health coverage and also considered a political challenge agenda (Ho et al., 2022; Ombere et al., 2023; Ombere”

  • The citation for Ombere is incomplete (missing the year).

Include a paragraph discussing the limitations of the study.

Author Response

Dear Author,

Below are the considerations for improving the writing of the manuscript.

Authorship:

The author states in the Funding section that “This article is based on my PhD dissertation” and in the Acknowledgments section presents the following information:
“I am also grateful to my PhD supervisors Prof. Dr. Tobias Haller (Institute of Social Anthropology, University of Bern, Switzerland) and Dr. Sonja Merten (Swiss Tropical and Public Health Institute, Switzerland).”
I request clarification on why the PhD supervisors are not listed as co-authors of the article.

Response: Thank you for the comment. My PhD supervisors are not listed as co-authors in this article but are on other articles (in-press) where they actively participated in crafting. But they are acknowledged in this one.

Keywords:

The keywords should be aligned with the MeSH thesaurus.
The following terms were not found in MeSH: Bottom-up; free maternity services; Kilifi County.

Response: Thank you, the terms not found in MeSH have been deleted from keywords
Replace “universal health coverage” with “Universal Health Care.”
Replace “qualitative” with “Qualitative Research.”

Response: Thank you these suggestions have been incorporated

Introduction:

In the sentence: “In Kenya today, UHC symbolizes a shift away from a health care system focused on the Market,” could you better explain what “Market” refers to?
See: “Universal Healthcare (UHC) agenda.”

Response: this has been revised to bring out the message vividly.

“UHC appears to represent a new approach and new ways of thinking about poverty and redistribution, the state and citizenship, health and development…..”

Response:
In the fifth paragraph of the introduction, I suggest explaining how the current healthcare system in Kenya is organized in general (including both primary care and hospital care), with a more detailed focus on maternal care.
Is the right to health in Kenya a constitutional right?
For policymakers in Kenya, what is their concept of health?
Include a contextualization of the Vision 2030 document – https://vision2030.go.ke/about-vision-2030/

Response: Thank you for this insightful comment and suggestion. I have now described the current healthcare system and how it is organized and included statements on health as a constitutional right and the link between UHC and Vision 2030 in Kenya.

Healthcare service in Kenya is provided by public health hospitals, private-for-profit facilities and non-governmental organizations. Public health facilities are organized around a four-level system: (1) community services, (2) primary health services, (3) county referral services and (4) national referral services (Tsofa et al., 2017; McCollum et al., 2018). Although both private and public facilities charge user fees to clients, fee is subsidized at level one and two in public hospitals. The private facilities are more expensive compared with public primarily serving wealthier individuals, whereas those from poorer households more commonly rely on public care providers or use lower standard, private care facilities (Ilinca et al., 2019). County governments are responsible for providing services in levels one to three, and national government is responsible for providing national referral services (Kairu et al., 2020; Masaba et al., 2020). In addition, under the new framework, responsibility for health service delivery is assigned to the counties, whereas policy, national referral hospitals and capacity building are the national government's responsibility (Masaba et al., 2020). Moreover, Kenya has enshrined the right to health in Article 43(1) (a) of the Constitution of Kenya, 2010, which provides that: “Every person has the right to the highest attainable standard of health, which includes the right to healthcare services, including reproductive health”(Owino et al., 2020). Additionally, Kenya's Vision 2030, a long-term development plan, acknowledges the importance of sexual and reproductive health rights (SRHR) as a key component of improving the quality of life for all Kenyans, particularly focusing on adolescents and young people. This is reflected in the Social Pillar of Vision 2030, which aims to provide equitable, affordable, and quality healthcare, including SRHR services through UHC (Oraro-Lawrence and Wyss, 2020; Ombere, 2024).

Objective:

The author needs to clearly define the objective of the study presented in this manuscript. Once defined, the objective should serve as the guiding thread of the article. The results should directly address this objective, and the conclusion should be strictly related to it and the data presented.

Response: Thank you for this suggestion. I have revised this in the manuscript and aligned results to address the objective.

The primary objective if this study entails

I suggest presenting the same objective at the end of the introduction and in the abstract. However, in my opinion, both are not well aligned with the study.
My suggested objective is:

explore health facility administrators’ views on whether intersectoral collaboration could help in the realization of UHC in Kenya
OR

Explain healthcare workers’ viewpoints on the possibilities of intersectoral collaborations in attaining UHC in Kenya

Response: Thank you for the comment: I have now added the primary objective of this article in the body and the abstract. “explore health facility administrators’ views on whether intersectoral collaboration could help in the realization of UHC in Kenya”

In the conclusion, the following statement appears:
“This study is a reflection on implementing the expanded free maternity services in Kenya dubbed Linda Mama.”

Response: Thank you for pointing this out. This has been revised and now reads “This study is a follow-up study on implementing the expanded free maternity services in Kenya, dubbed Linda Mama.

Methods:

The Methods section should be aligned with the COREQ checklist (See EQUATOR Network – https://www.equator-network.org/reporting-guidelines/coreq/). All items should be covered in the methods section.

Item 2.3

Study Sample:

This section requires substantial revisions to clearly describe who the study participants were and how they were recruited.

  • The author must clearly state who participated in the research described in this manuscript.
  • Who were the participants in the individual interviews?
  • Since this is a longitudinal study, did the same participants take part in both the first phase (July 2016 and January–July 2017) and the second phase (2020 and 2021)? Which months were covered in the second phase?

Response: Thank you for the comments. I have now revised in methodology who participated in the study and how they were selected. Moreover, the second phase of this study covered (June–July 2020 and September–October 2020). This is also now added in the manuscript. Yes, the same health administrators who had participated in the first phase and all of them were available to participate during second phase took part in this study.

  • Who participated in the two round-table discussions? How were these participants selected?

Response: Thank you, the participants were health administrators. They were selected on their availability and willingness to participate during the study. Thus, this was purposive as highlighted now in the manuscript.

  • Did the individuals who took part in the informal conversations give consent to participate in the research? Were they aware that the information from these conversations would be used in a scientific study? How was the data from these conversations recorded?

Response: thank you for the comment. The informal interviews were conducted between July 2016 and January–July 2017. Yes, the participants were aware since I spent time in the referral hospitals and had introduced myself and nature of research I was doing.

“The researcher purposively picked five matrons—nurse-midwives who oversee departments—during my first fieldwork and also selected the two busiest health facilities.”

  • What criteria guided this purposive selection?

Thank you for this comment, the nurses-midwives were purposively picked based on the inclusion criteria in this study. Now highlighted in the document “key healthcare officials dealing with maternal and child health and also facilitating health issues in the hospital and at the county level. “

“The five matrons that had initially been interviewed formed the study population as described in the author’s previous study.”

  • Are the matrons the study population for this research? This is unclear.

Response: Thank you. The study population included the matrons and the health administrators from the county. Now added in the manuscript “The study population for this study thus was the matrons and the health administrators in the county.”

Methods of Data Collection:

  • Provide information about the interview and round-table discussion guides.
  • What tool was used to record the data collection?

Response: Thank you. The information is now added in the manuscript “During the discussion, notes were taken and the discussion recorded using audio-voice recorder. “

2.4. Data Analysis:

  • “The conclusion of the data analysis was reached when no new themes emerged.”
    • Is this correct? Or was it the conclusion of data collection?

Response: thank you for this comment. This has now been corrected in the manuscript “Data analysis stopped when there were no new themes emerging.”

2.5. Ethical Procedure:

  • Ethical approval was obtained from Maseno University Ethical Review Committee, reference number MSU/DRPI/MUERC/00206/015, date of approval: March 2, 2017.
  • However, data collection started in July 2016, before ethical approval was granted. Could you clarify this issue?

Response: Thank you for this comment since this study was based on a multidisciplinary project, part of it started in 2016 where we had an ethical approval  and due to change in research approach for the larger project, we obtained another ethical approval in 2017.

Findings:

According to the COREQ guidelines, qualitative research results should be presented as an interpretation of the meanings attributed by the participants, supported by excerpts from participants’ statements that validate these interpretations.
Additionally, the study should include a characterization of the participants.
Thus, I suggest rewriting this section.

Response: Thank you. I have highlighted excerpts from the participants’ statements to validate the interpretations in the findings section.

Discussion:

It is necessary to include a discussion related to the Vision 2030 document – https://vision2030.go.ke/about-vision-2030/

Response: Thank you for pointing out this. After careful consideration, this has been corrected and added in the discussion section paragraph three.

“Moreover, despite the Kenya Vision 2030's social pillar emphasis on investing in people's well-being, with health as a key sector, findings from this study shows that affordable and accessible healthcare can be possible also through intersectoral collaborations. Such collaborations could help make healthcare services affordable and accessible to all citizens, regardless of their socio-economic status.”

Regarding this citation: “Studies have also shown that the free maternity services programme in Kenya was hurriedly launched, and it has been used as a pathway to universal health coverage and also considered a political challenge agenda (Ho et al., 2022; Ombere et al., 2023; Ombere”

  • The citation for Ombere is incomplete (missing the year).

Response: Thank you. This has been corrected.

Include a paragraph discussing the limitations of the study

Response: Thank you. This has been inserted and is now read.

Limitations

The main strength of this study is that this is one of the first anthropological studies conducted among health administrators to give a primary account of what can be done to enhance intersectoral collaboration in the expanded free maternity-Linda mama for the realization of universal healthcare. One of the major limitations is that the study only relied on health administrators’ perspectives and since they are policymakers at the county, some of the responses might have been exaggerated. However, to counter this, this study utilized varying data collection methods to triangulate the findings thus strengthening the findings. Moreover, the article lacks the perspectives of the administrators from other institutions such as non-governmental organizations, faith-based organizations, private hospital and international organizations dealing with maternal healthcare. This is a gap that can be researched on in future. While the findings may not be generalizable beyond Kilifi County because of the heterogeneity of the counties, this study identifies significant contextual factors that can be informative to policymakers as a guide to effective evidence-based interventions that can be adopted to strengthen the implementation of universal health coverage in Kenya

Reviewer 3 Report

Comments and Suggestions for Authors

Dear Authors,

The present study is an extension of the authors' previous study (Ombere et al., 2023; Ombere, 2024). The authors should engage the reader with the present study by clarifying the research strategy and its results. 

It is suggested that the focus should be on the core of the manuscript and the rationality of its objectives, as opposed to general considerations of Universal Health Coverage (UHC). This paper is about maternal health as an approach to UHC in Kenya. Is this an accurate interpretation? If so, it should be presented and detailed, and the authors should explain the FMS.

The study design would be improved by including a diagram or picture.
What was the research strategy?
Who was the target population?
The authors stated: "In addition, the author's handwritten notes from informal conversations were manually coded and added to the study...The author and his supervisors evaluated the codes to identify recurring themes throughout the study".This process should be explained in more detail. 
The results section is difficult to follow; the authors should present the results schematically.
The conclusions section is difficult to follow, particularly concerning the Linda Mama project.  

Comments on the Quality of English Language

The present study is an extension of the authors' previous study (Ombere et al., 2023; Ombere, 2024). The authors should engage the reader with the present study by clarifying the research strategy and its results. 

It is suggested that the focus should be on the core of the manuscript and the rationality of its objectives, as opposed to general considerations of Universal Health Coverage (UHC). This paper is about maternal health as an approach to UHC in Kenya. Is this an accurate interpretation? If so, it should be presented and detailed, and the authors should explain the FMS.

The study design would be improved by including a diagram or picture.
What was the research strategy?
Who was the target population?
The authors stated: "In addition, the author's handwritten notes from informal conversations were manually coded and added to the study...The author and his supervisors evaluated the codes to identify recurring themes throughout the study".This process should be explained in more detail. 
The results section is difficult to follow; the authors should present the results schematically.
The conclusions section is difficult to follow, particularly concerning the Linda Mama project.  

Author Response

Dear author,

The present study is an extension of the authors' previous study (Ombere et al., 2023; Ombere, 2024). The authors should engage the reader with the present study by clarifying the research strategy and its results. 

Response: Thank you, despite being a follow-up study, I have now revised methodology to engage readers on the present study and its relevance in UHC.

It is suggested that the focus should be on the core of the manuscript and the rationality of its objectives, as opposed to general considerations of Universal Health Coverage (UHC). This paper is about maternal health as an approach to UHC in Kenya. Is this an accurate interpretation? If so, it should be presented and detailed, and the authors should explain the FMS.

Response: Thank you. However, the paper largely talks about intersectoral collaboration through maternal healthcare for attainment of UHC in Kenya. I have revised sections in the introduction to make this clearer.

The study design would be improved by including a diagram or picture.
What was the research strategy?
Who was the target population?
The authors stated: "In addition, the author's handwritten notes from informal conversations were manually coded and added to the study...The author and his supervisors evaluated the codes to identify recurring themes throughout the study".This process should be explained in more detail. 

Response: Thank you, after careful consideration. I have decided not to include diagram or picture in this qualitative study instead, I have added information in the methodology that makes the design, research strategy, target population, and data analysis more clearer.

The results section is difficult to follow; the authors should present the results schematically.
The conclusions section is difficult to follow, particularly concerning the Linda Mama project.  

Response: thank you for the comment. After careful considerations, I have adjusted discussion and conclusion sections to clearly bring out issues on Linda Mama which is an expanded free maternity programme but not a project.

Round 2

Reviewer 2 Report

Comments and Suggestions for Authors

Abstract – I suggest rewriting it. The background section is excessively long. Remove the following sentence: “This article explores the possibilities of intersectoral collaboration, specifically in maternal healthcare, and what can be done to realize such collaborations to drive universal health coverage (UHC) in Kenya” as it is already stated in the objective.

Methods – Include the research setting and review the study period (the main text states: “conducted between June–July 2020 and September–October 2020”). The study participants should be explicitly indicated (“The study population for this study thus was the matrons and the health administrators in the county”). Regarding the study type, I suggest replacing the term “follow-up” in the phrase “The article is based on a follow-up qualitative research” with “longitudinal”.

Review the objective in both the abstract and the main text – The phrase should include the matrons.

Introduction

I suggest revising the sentence: “In the SDGs, there is only one goal explicitly talking about health (SDG 3). Moreover, some targets are important to health improvement across the other 16 SDGs [46]”, as the WHO recommends working with the broader concept of health, which considers social determinants of health and, consequently, the need for intersectoral collaboration.

Methods

2.2 Study Design

Remove the sentence: “The study explored health facility administrators’ views on whether intersectoral collaboration could help in the realization of UHC in Kenya.” I suggest making the study type description more concise by explicitly stating that it is longitudinal qualitative research and including a definition of this study type along with a supporting reference.

2.3 Study Sample and Methods of Data Collection

The writing in this section is not entirely clear for the reviewer.

To clarify, please describe Study Sample under section 2.3:

  • Who participated in the study (“The study population for this study thus was the matrons and the health administrators in the county”).

  • How these participants were selected – clarify that the participants were the same as those from data collection conducted from March to July 2016 and also from January to July 2017.

  • How participants were invited to take part in the second phase of the study (see item 11 COREQ guideline).

Include a new section: 2.4 Methods of Data Collection

  • Here, describe the individual interviews (and specify who participated), informal conversations, and round-table discussions.

  • Provide details about the interview and round-table discussion guides (see item 17 COREQ guideline).

The information given in the first round about this question must be included in the text:

  • Did the individuals who took part in the informal conversations give consent to participate in the research? Were they aware that the information from these conversations would be used in a scientific study? How was the data from these conversations recorded?

Response: Thank you for the comment. The informal interviews were conducted between July 2016 and January–July 2017. Yes, the participants were aware since I spent time in the referral hospitals and had introduced myself and the nature of the research I was conducting.

Include information from items 21 and 23 of COREQ.

Ethical Procedure:

Include the ethical approval from the multidisciplinary project, part of which started in 2016.

Results

  • Standardize the themes mentioned in the phrase: “These themes include fair representation of stakeholders, the need for collaborative and distributed leadership, and local participation and bargaining power” to match the titles of the presented subtopics.

  • Are there more findings to share with readers? Since this is a longitudinal qualitative study, I expected to find interpretations from both the first and second data collections, and in the discussion, you should argue whether there were differences.

  • The reporting of research findings must respond to the study objective and the adopted method.

Discussion

  • Ensure accuracy when referring to participant groups in sentences like: “This section discusses key themes from the varying views of health facility administrators and county health officials regarding enhancing intersectoral collaborations in maternal healthcare to realize universal health coverage in Kenya.” The matrons must be included! Please revise throughout the text.

  • The mention of matrons is also absent in the study limitations; this must be corrected.

Author Response

Dear Reviewer,

I am grateful for your insights and comments, which have helped me improve the manuscript. I have pasted the responses here and highlighted them in red font in the main manuscript.

Thank you

Authorship:

The author states in the Funding section that “This article is based on my PhD dissertation” and in the Acknowledgments section presents the following information:
“I am also grateful to my PhD supervisors Prof. Dr. Tobias Haller (Institute of Social Anthropology, University of Bern, Switzerland) and Dr. Sonja Merten (Swiss Tropical and Public Health Institute, Switzerland).”
I request clarification on why the PhD supervisors are not listed as co-authors of the article.

Response: Thank you for the comment. My PhD supervisors are not listed as co-authors in this article but are on other articles (in-press) where they actively participated in crafting. But they are acknowledged in this one.

Keywords:

The keywords should be aligned with the MeSH thesaurus.
The following terms were not found in MeSH: Bottom-up; free maternity services; Kilifi County.

Response: Thank you, the terms not found in MeSH have been deleted from keywords
Replace “universal health coverage” with “Universal Health Care.”
Replace “qualitative” with “Qualitative Research.”

Response: Thank you these suggestions have been incorporated

Introduction:

In the sentence: “In Kenya today, UHC symbolizes a shift away from a health care system focused on the Market,” could you better explain what “Market” refers to?
See: “Universal Healthcare (UHC) agenda.”

Response: this has been revised to bring out the message vividly.

“UHC appears to represent a new approach and new ways of thinking about poverty and redistribution, the state and citizenship, health and development…..”

Response:
In the fifth paragraph of the introduction, I suggest explaining how the current healthcare system in Kenya is organized in general (including both primary care and hospital care), with a more detailed focus on maternal care.
Is the right to health in Kenya a constitutional right?
For policymakers in Kenya, what is their concept of health?
Include a contextualization of the Vision 2030 document – https://vision2030.go.ke/about-vision-2030/

Response: Thank you for this insightful comment and suggestion. I have now described the current healthcare system and how it is organized and included statements on health as a constitutional right and the link between UHC and Vision 2030 in Kenya.

Healthcare service in Kenya is provided by public health hospitals, private-for-profit facilities and non-governmental organizations. Public health facilities are organized around a four-level system: (1) community services, (2) primary health services, (3) county referral services and (4) national referral services (Tsofa et al., 2017; McCollum et al., 2018). Although both private and public facilities charge user fees to clients, fee is subsidized at level one and two in public hospitals. The private facilities are more expensive compared with public primarily serving wealthier individuals, whereas those from poorer households more commonly rely on public care providers or use lower standard, private care facilities (Ilinca et al., 2019). County governments are responsible for providing services in levels one to three, and national government is responsible for providing national referral services (Kairu et al., 2020; Masaba et al., 2020). In addition, under the new framework, responsibility for health service delivery is assigned to the counties, whereas policy, national referral hospitals and capacity building are the national government's responsibility (Masaba et al., 2020). Moreover, Kenya has enshrined the right to health in Article 43(1) (a) of the Constitution of Kenya, 2010, which provides that: “Every person has the right to the highest attainable standard of health, which includes the right to healthcare services, including reproductive health”(Owino et al., 2020). Additionally, Kenya's Vision 2030, a long-term development plan, acknowledges the importance of sexual and reproductive health rights (SRHR) as a key component of improving the quality of life for all Kenyans, particularly focusing on adolescents and young people. This is reflected in the Social Pillar of Vision 2030, which aims to provide equitable, affordable, and quality healthcare, including SRHR services through UHC (Oraro-Lawrence and Wyss, 2020; Ombere, 2024).

Objective:

The author needs to clearly define the objective of the study presented in this manuscript. Once defined, the objective should serve as the guiding thread of the article. The results should directly address this objective, and the conclusion should be strictly related to it and the data presented.

Response: Thank you for this suggestion. I have revised this in the manuscript and aligned results to address the objective.

The primary objective if this study entails

I suggest presenting the same objective at the end of the introduction and in the abstract. However, in my opinion, both are not well aligned with the study.
My suggested objective is:

explore health facility administrators’ views on whether intersectoral collaboration could help in the realization of UHC in Kenya
OR

Explain healthcare workers’ viewpoints on the possibilities of intersectoral collaborations in attaining UHC in Kenya

Response: Thank you for the comment: I have now added the primary objective of this article in the body and the abstract. “explore health facility administrators’ views on whether intersectoral collaboration could help in the realization of UHC in Kenya”

In the conclusion, the following statement appears:
“This study is a reflection on implementing the expanded free maternity services in Kenya dubbed Linda Mama.”

Response: Thank you for pointing this out. This has been revised and now reads “This study is a follow-up study on implementing the expanded free maternity services in Kenya, dubbed Linda Mama.

Methods:

The Methods section should be aligned with the COREQ checklist (See EQUATOR Network – https://www.equator-network.org/reporting-guidelines/coreq/). All items should be covered in the methods section.

Item 2.3

Study Sample:

This section requires substantial revisions to clearly describe who the study participants were and how they were recruited.

  • The author must clearly state who participated in the research described in this manuscript.
  • Who were the participants in the individual interviews?
  • Since this is a longitudinal study, did the same participants take part in both the first phase (July 2016 and January–July 2017) and the second phase (2020 and 2021)? Which months were covered in the second phase?

Response: Thank you for the comments. I have now revised the methodology of who participated in the study and how they were selected. Moreover, the second phase of this study covered (June–July 2020 and September–October 2020). This is also now added in the manuscript. Yes, the same health administrators who had participated in the first phase and all of them were available to participate during second phase took part in this study.

  • Who participated in the two round-table discussions? How were these participants selected?

Response: Thank you, the participants were health administrators. They were selected on their availability and willingness to participate during the study. Thus, this was purposive as highlighted now in the manuscript.

  • Did the individuals who took part in the informal conversations give consent to participate in the research? Were they aware that the information from these conversations would be used in a scientific study? How was the data from these conversations recorded?

Response: thank you for the comment. The informal interviews were conducted between July 2016 and January–July 2017. Yes, the participants were aware since I spent time in the referral hospitals and had introduced myself and nature of research I was doing.

“The researcher purposively picked five matrons—nurse-midwives who oversee departments—during my first fieldwork and also selected the two busiest health facilities.”

  • What criteria guided this purposive selection?

Thank you for this comment, the nurses-midwives were purposively picked based on the inclusion criteria in this study. Now highlighted in the document “key healthcare officials dealing with maternal and child health and also facilitating health issues in the hospital and at the county level. “

“The five matrons that had initially been interviewed formed the study population as described in the author’s previous study.”

  • Are the matrons the study population for this research? This is unclear.

Response: Thank you. The study population included the matrons and the health administrators from the county. Now added in the manuscript “The study population for this study thus was the matrons and the health administrators in the county.”

Methods of Data Collection:

  • Provide information about the interview and round-table discussion guides.
  • What tool was used to record the data collection?

Response: Thank you. The information is now added in the manuscript “During the discussion, notes were taken and the discussion recorded using audio-voice recorder. “

2.4. Data Analysis:

  • “The conclusion of the data analysis was reached when no new themes emerged.”
    • Is this correct? Or was it the conclusion of data collection?

Response: thank you for this comment. This has now been corrected in the manuscript “Data analysis stopped when there were no new themes emerging.”

2.5. Ethical Procedure:

  • Ethical approval was obtained from Maseno University Ethical Review Committee, reference number MSU/DRPI/MUERC/00206/015, date of approval: March 2, 2017.
  • However, data collection started in July 2016, before ethical approval was granted. Could you clarify this issue?

Response: Thank you for this comment since this study was based on a multidisciplinary project, part of it started in 2016 where we had an ethical approval  and due to change in research approach for the larger project, we obtained another ethical approval in 2017.

Findings:

According to the COREQ guidelines, qualitative research results should be presented as an interpretation of the meanings attributed by the participants, supported by excerpts from participants’ statements that validate these interpretations.
Additionally, the study should include a characterization of the participants.
Thus, I suggest rewriting this section.

Response: Thank you. I have highlighted excerpts from the participants’ statements to validate the interpretations in the findings section.

Discussion:

It is necessary to include a discussion related to the Vision 2030 document – https://vision2030.go.ke/about-vision-2030/

Response: Thank you for pointing out this. After careful consideration, this has been corrected and added in the discussion section paragraph three.

“Moreover, despite the Kenya Vision 2030's social pillar emphasis on investing in people's well-being, with health as a key sector, findings from this study shows that affordable and accessible healthcare can be possible also through intersectoral collaborations. Such collaborations could help make healthcare services affordable and accessible to all citizens, regardless of their socio-economic status.”

Regarding this citation: “Studies have also shown that the free maternity services programme in Kenya was hurriedly launched, and it has been used as a pathway to universal health coverage and also considered a political challenge agenda (Ho et al., 2022; Ombere et al., 2023; Ombere”

  • The citation for Ombere is incomplete (missing the year).

Response: Thank you. This has been corrected.

Include a paragraph discussing the limitations of the study

Response: Thank you. This has been inserted and is now read.

Limitations

The main strength of this study is that this is one of the first anthropological studies conducted among health administrators to give a primary account of what can be done to enhance intersectoral collaboration in the expanded free maternity-Linda mama for the realization of universal healthcare. One of the major limitations is that the study only relied on health administrators’ perspectives and since they are policymakers at the county, some of the responses might have been exaggerated. However, to counter this, this study utilized varying data collection methods to triangulate the findings thus strengthening the findings. Moreover, the article lacks the perspectives of the administrators from other institutions such as non-governmental organizations, faith-based organizations, private hospital and international organizations dealing with maternal healthcare. This is a gap that can be researched on in future. While the findings may not be generalizable beyond Kilifi County because of the heterogeneity of the counties, this study identifies significant contextual factors that can be informative to policymakers as a guide to effective evidence-based interventions that can be adopted to strengthen the implementation of universal health coverage in Kenya

Reviewer 3 Report

Comments and Suggestions for Authors

The authors have responded positively to the suggestions of the review process.

Comments on the Quality of English Language

The authors have responded positively to the suggestions of the review process.

Author Response

Dear Reviewer,

I am grateful for your insights and comments which have immensely shaped and improved my article. Please find my point-to-point responses. I have highlighted them using red font colour in the manuscript.

Thank you

The present study is an extension of the authors' previous study (Ombere et al., 2023; Ombere, 2024). The authors should engage the reader with the present study by clarifying the research strategy and its results. 

Response: Thank you, despite being a follow-up study, I have now revised methodology to engage readers on the present study and its relevance in UHC.

It is suggested that the focus should be on the core of the manuscript and the rationality of its objectives, as opposed to general considerations of Universal Health Coverage (UHC). This paper is about maternal health as an approach to UHC in Kenya. Is this an accurate interpretation? If so, it should be presented and detailed, and the authors should explain the FMS.

Response: Thank you. However, the paper largely talks about intersectoral collaboration through maternal healthcare for attainment of UHC in Kenya. I have revised sections in the introduction to make this clearer.

The study design would be improved by including a diagram or picture.
What was the research strategy?
Who was the target population?
The authors stated: "In addition, the author's handwritten notes from informal conversations were manually coded and added to the study...The author and his supervisors evaluated the codes to identify recurring themes throughout the study".This process should be explained in more detail. 

Response: Thank you, after careful consideration. I have decided not to include diagram or picture in this qualitative study instead, I have added information in the methodology that makes the design, research strategy, target population, and data analysis more clearer.

The results section is difficult to follow; the authors should present the results schematically.
The conclusions section is difficult to follow, particularly concerning the Linda Mama project.  

Response: thank you for the comment. After careful considerations, I have adjusted discussion and conclusion sections to clearly bring out issues on Linda Mama which is an expanded free maternity programme but not a project.